# Event detection in football: Improving the reliability of match analysis

**Jonas Bischofberger**[1]*, **Arnold Baca**[1], **Erich Schikuta**[2]

**1** Centre for Sport Science and University Sports, University of Vienna, Vienna, Austria, **2** Faculty of Computer Science, University of Vienna, Vienna, Austria

* jonas.bischofberger@univie.ac.at

**Data Availability Statement:** The data underlying this research is partially available under the following URL: https://github.com/metrica-sports/sample-data The remaining data comes from

## Abstract

With recent technological advancements, quantitative analysis has become an increasingly important area within professional sports. However, the manual process of collecting data on relevant match events like passes, goals and tacklings comes with considerable costs and limited consistency across providers, affecting both research and practice. In football, while automatic detection of events from positional data of the players and the ball could alleviate these issues, it is not entirely clear what accuracy current state-of-the-art methods realistically achieve because there is a lack of high-quality validations on realistic and diverse data sets. This paper adds context to existing research by validating a two-step rule-based pass and shot detection algorithm on four different data sets using a comprehensive validation routine that accounts for the temporal, hierarchical and imbalanced nature of the task. Our evaluation shows that pass and shot detection performance is highly dependent on the specifics of the data set. In accordance with previous studies, we achieve F-scores of up to 0.92 for passes, but only when there is an inherent dependency between event and positional data. We find a significantly lower accuracy with F-scores of 0.71 for passes and 0.65 for shots if event and positional data are independent. This result, together with a critical evaluation of existing methodologies, suggests that the accuracy of current football event detection algorithms operating on positional data is currently overestimated. Further analysis reveals that the temporal extraction of passes and shots from positional data poses the main challenge for rule-based approaches. Our results further indicate that the classification of plays into shots and passes is a relatively straightforward task, achieving F-scores between 0.83 to 0.91 ro rule-based classifiers and up to 0.95 for machine learning classifiers. We show that there exist simple classifiers that accurately differentiate shots from passes in different data sets using a low number of human-understandable rules. Operating on basic spatial features, our classifiers provide a simple, objective event definition that can be used as a foundation for more reliable event-based match analysis.

## 1. Introduction

The objective evaluation of performance is a ubiquitous goal in modern professional sports environments. When recruiting players or analyzing opponents, it is crucial to be able to assess

commercial sources and we are not allowed to share it. It can be requested from the Austrian Football Federation (https://www.oefb.at/), via the "Abteilung für Wissenschaft, Analyse und Entwicklung" or directly from the various data providers/rights holders, i.e. Stats Perform (https://www.statsperform.com/), the UEFA (https://www.uefa.com/), and Subsequent (https://subsequent.ai/).

**Funding:** The authors received no specific funding for this work.

**Competing interests:** The authors have declared that no competing interests exist.

the overall capabilities as well as strengths and weaknesses of teams and athletes. Furthermore, objective, quantitative analysis has the power to reduce the impact of individual and societal biases which could ultimately lead to more truthful and healthy relationships between athletes, coaches, and the public.

With growing sophistication and decreasing costs, technologies such as video-based systems, electronic tracking systems and match analysis software become more and more widespread, leading to an increasingly important role of quantitative analysis in sports. In football, tactical and technical performance analysis traditionally focuses on player actions such as shots, passes and dribbles [1]. The data for these analyses is collected through manual tagging of events which is a time-consuming and cost-intensive process. Additionally, while the reliability of manual event detection systems can be ensured by extensive training of the data collectors [2], their validity is harder to guarantee. Data providers are not required to publish accurate and detailed definitions of the events they annotate. Definitions also vary across providers, because most football concepts are not prescribed by the rules of the game but emerged empirically, which makes their definitions subject to opinion. For example, the term *recovery* is used for very different sets of actions, ranging from a player picking up a loose ball [3] to any action by which a player wins the ball [4]. Even foundational actions such as passes, dribbles and shots are typically ambiguous: For example, the provider *Wyscout* treats crosses as a subset of passes whereas *Stats Perform* does not. *Stats Perform* also requires a pass to be *intentional*, which is hardly an objective qualificiation.

Without universal definitions, different studies which seemingly use the same categories of events are not necessarily comparable. Also, the event definition that is required or expected by an analyst or researcher might differ from the definition used by the data collector. For that reason, an automated and customizable data collection process would increase the validty of both scientific and practiced sports performance analysis—if such a process is sufficiently accurate.

So far, various methods to automatically extract events from raw video footage or positional data of the players and the ball have been proposed, using either machine learning or rule-based detection routines. A rule-based algorithm operating on positional data would be particularly well suited to not only alleviate the burden of manual data collection, but also provide a simple, objective definition of events as a foundation for further analysis. Multiple machine-learning- and rule-based methods have been proposed to detect events in positional data, reporting promising accuracies of 90% and above [5–8]. However, most studies did not evaluate their algorithms across multiple data sets, so it is not guaranteed that these algorithms pick up the underlying structure of the football game rather than the error profile or other specifics of the respective data set. Also, and more importantly, the data sets that were used for validation are typically not independent from the positional data, as they both come from a common intermediate source or are partially derived from each other. Using such data for the evaluation of an algorithm inevitably leads to an inflation of its estimated performance, since information from the reference data spills over into the input data for the model.

This article complements and enriches those previous findings by providing a strong validation of a simple rule-based algorithm for the detection of passes and shots as two of the most important events in football from positional data. We propose a highly robust validation routine and use it to evaluate the algorithm across four different data sets, where one data set includes independent positional and event data. We also compare different algorithms to further distinguish passes and shots, including both rule-based and machine-learning classifiers to determine whether there exists a simple, human-understandable set of rules which accurately distinguishes shots from passes.

Designing a proper validation routine for this problem is technically challenging, because it involves detecting composite events from a continuous stream of positional data. It is a temporal, hierarchical and imbalanced classification task with unreliable reference data. The suggested validation routine is therefore relevant beyond the scope of football event detection for problems with a similar structure, such as object detection from videos [9] or sentiment analysis from streams of social media posts [10].

Overall, the main novel contributions of this paper are:

- The presentation of a reliable validation routine for football event detection as a temporal, hierarchical, and imbalanced classification problem.

- A reliable estimate of the performance of different pass and shot detection algorithms based on positional data across four diverse data sets.

- A quantification and exploration of the difference in performance between independent and dependent reference data, which adds important context to existing findings.

- An accurate pass and shot classifier that can be used as an adjustable foundation for event-based match analysis.

The remaining paper is structured as follows: Section 2 reviews existing approaches to automatic event detection in football. Section 3 elaborates the pass and shot detection algorithms evaluated in this paper. Section 4 describes the data sets used and lays out the design of the validation procedure. Section 5 presents the validation results. Section 6 provides a discussion of the results. Section 7 summarizes the paper and proposes directions for future research.

## 2. State of the art

In football, there are different types of events that are relevant for performance analysis: Player-level actions such as runs, tackles and passes, team- or group-level events such as counterattacks, offside traps and changes of the tactical formation, and events prescribed by the rules of football, for instance game interruptions when the ball leaves the bounds of the pitch and substitutions. Player actions form the building blocks upon which the majority of technical and tactical performance analysis in football is built, such as the analysis of pass completion rates [1], expected goals [11] and passing networks [12].

Event recognition from sports videos is an active area of research. It can involve either machine learning [13] or basic image recognition techniques in conjunction with manual classification rules [14, 15]. Given that positional data becomes more and more widely available and that many events can be defined in terms of spatio-temporal interactions between players and the ball, it becomes increasingly feasible to perform event detection on positional data on a large scale. In fact, some approaches in video-based event detection even involve the recognition of players and the ball as a preprocessing step [14].

One of the earliest attempts to rule-based event detection on positional data in football can be attributed to Tovinkere and Qian [8] who used rules and heuristics derived from domain knowledge to identify complex exents in positional data. They achieved F-scores very close or equal to 1.0 for kicks, receptions, saves and goals. However, details about their methods are too sparsely given to draw general conclusions from this result. Also, they evaluated their algorithm on a very small sample which contained only 101 player actions.

More recently, Morra et al. [6] proposed and evaluated a rule-based algorithm on positional data, which is able to extract complex events like passes from atomic events using sets of temporal and logical rules. The achieved F-scores of 0.89 (passes) and 0.81 (shots) are likely inflated by the use of synthetic positional and reference event data, which have been jointly

generated from a football simulation engine. Khaustov and Mozgovoy [7] applied another rule-based algorithm to positional data from 5 football matches and achieved an F-score of 0.93 for successful passes, 0.86 for unsuccessful passes, and 0.95 for shots on goal. Even higher values were achieved, such as 0.998 for successful passes, on another data set containing several short sequences of play. However, they also generated their gold standard by hand by watching the game "in a 2D soccer simulator", i.e. likely using the same positional data that also underlies the event detection process.

Richly, Moritz and Schwarz [16] used a three-layer artificial neural network to detect events in positional data and achieved an average F-score of 0.89 for kicks and receptions. However, they also used positional data to assist the manual creation of their gold standard, specifically the acceleration data of the ball. They also used a very small data set with only 194 events in total. Vidal-Codina et al. [5] proposed a two-step rule-based detector and evaluated it on a very heterogeneous data set, however with no discussion of possible data dependencies and differences of the algorithm's performance between the various included providers. Among other events, they achieved a total F-score of 0.92 for passes and 0.63 for shots.

Overall, while the achieved F-scores beyond 90% for passes and shots appear promising, the currently available evaluation results don't necessarily reflect a practical setting where manual event data is supposed to be replaced and is therefore not available to pre-process positional data. It is likely that the existing studies tend to overestimate the accuracy of their algorithms due to information from the reference data leaking into the input data. For that reason, it is currently not clear which merit rule-based classification routines hold concerning event detection in football and if their accuracy is sufficient for industrial and research purposes. Also, there is a lack of agreed-upon standards regarding the validation a given algorithm, for example the specific conditions under which a detected event can be considered to match a reference event. Other problems like low sample sizes, a lack of variety in the evaluated data sets and the use of synthetic data further emphasize the need for new perspectives on the topic.

## 3. Approach to detection and classification

We propose a simple rule-based algorithm using positional data to detect passes and shots, as two of the most important and widely analyzed actions in football. The structure of the algorithm is hierarchical as passes and shots can both be viewed as actions where the ball is kicked by a player. Therefore, in a first step, *plays* (defined as actions that are either a pass or a shot) are detected from the positional data. In the second step, plays are classified into passes and shots using three different methods: A rule-based decision routine based on expert knowledge, decision trees of various complexity, and various black-box machine learning classifiers. The rule-based routine and decision trees are used to estimate the achievable performance of a human-understandable classifier which is desirable to obtain event defintions. The black box classifiers are used to estimate whether a higher accuracy is possible without imposing this requirement.

All models have been implemented in Python 3.9 and rely on the packages numpy (1.21.5), pandas (1.4.4) and scikit-learn (1.0.2).

### Step 1: Play detection

The basic idea of detecting passes and shots is to view them as composed of two actions where a player exerts a force upon the ball, i.e. a kick followed by either a reception or deflection. Parsimoniously, a *hit* is defined as an instance where the ball is accelerated beyond a certain threshold (`min_acc`) while at least one player is in sufficient vicinity of the ball (`min_vicinity`) to have realistically carried out the hit. The player performing the hit is

defined as the closest player to the ball during the hit. With the acceleration $a_{Ball}(f)$ of the ball in frame $f$ and the minimal ball-player distance over all players $d_{closest}(f)$, the occurence of a hit in $f$ is defined as:

$$Hit(f) := a_{Ball}(f) > \mathtt{min\_acc} \text{ and } d_{closest}(f) < \mathtt{min\_vicinity} \tag{1}$$

A play is then defined as either two subsequent hits by different players (which corresponds to a pass or shot followed by a reception, deflection or goalkeeper save) or a hit followed by a game interruption (e.g. a shot that misses the target or a misplaced pass that crosses the sideline).

This definition of a play is broad and captures all "pass-like" events such as crosses, deflections, clearances, misplaced touches and throw-ins which may or may not be considered passes in different contexts. If one wants to further subdivide the pass event into categories, this can be done explicitly through additional rules. For example, a cross could be defined as a pass that originates in a specified zone lateral to the penalty box, is received in the penalty box, and reaches a certain height during its trajectory.

Since crosses, clearances and throw-ins are commonly recorded in football event data, we include those events as passes in the evaluation. Deflections and misplaced touches on the other hand are not always recorded, so they should be excluded algorithmically. Misplaced touches are difficult to detect because essentially, misplaced touches differ from passes by the intention of the player rather than directly observable parameters of the play. While the same is true for deflections, deflections appear to have the more distinct kinematic features. Intuitively, deflections can be thought of as plays that directly follow another play when the deflecting player has not had sufficient time to make a conscious play. We therefore use the following rule to classify plays as deflections and exclude those from the final set of detected plays.

$$Deflection(play) := player.frame = previous\_play.target\_frame \text{ and}$$
$$previous\_play.duration < \mathtt{max\_deflection\_time} \tag{2}$$

Algorithm 1 describes the procedure programmatically. Given that all calculations per frame run in constant time, its time complexity is $O(n)$ where $n$ is the number of frames in the positional data. The space complexity is also $O(n)$.

**Algorithm 1** Play Detection Algorithm

```
 1: function ISDEFLECTION(play, previousPlay)
 2:   return play.frame = previousPlay.target_frame and previousPlay.
    duration < max_deflection_time
 3:
 4: function ISHIT(f, min_acc, min_vicinity)
 5:   return a_Ball(f) > min_acc and d_closest(f) < min_vicinity
 6:
 7: function DETECTPLAYS(game, min_acc, min_vicinity, max_deflection_time)
 8:   plays ← []
 9:   startFrame ← -1
10:   firstHit ← -1
11:   previousClosestPlayer ← -1
12:   previousPlay ← -1
13:   for each frame f in game do
14:     closestPlayer ← CLOSESTPLAYERTOBALLF(f)
15:     if ISHIT(f, min_acc, min_vicinity) and previousClosestPlayer ≠
    closestPlayer then
16:       if firstHit = -1 then
17:         firstHit ← f
18:       else if note ISDEFLECTION(play, previousPlay) then
```

```
19:        Append (firstHit, f) to plays
20:        previousPlay ← (firstHit, f)
21:        firstHit ← -1
22:    else if f is Interruption and firstHit ≠ -1 then
23:      Append (firstHit, f) to plays
24:      previousPlay ← (firstHit, f)
25:      firstHit ← -1
26: return Detected plays
```

## Step 2: Shot classification

To classify the detected plays into passes and shots, we first extract a manually defined set of features. The features, are shown in Table 1 and are selected because they seem to be both simple and informative for differentiating shots from passes. These features serve as input for all classifiers.

**Manual algorithm.** For the first algorithm, we use the extracted features to carefully build a set of rules that classifies plays as shots or passes. These rules are designed to capture expert intuition about what constsitutes a shot in football. A play is therefore classified as a shot if and only if it satisfies all of the following rules:

1. *play.progressed_dist_toward_goal* > `min_progression`: A shot has to bring the ball closer to the opponent's goal.

2. *play.origin.distance_to_goal* < `max_dist_to_goal`: A shot has to be taken within sufficient vicinity of the opponent's goal.

3. *play.origin.opening_angle_to_goal* < `min_opening_angle` **or** *play.initial_speed* > `min_speed_from_bad_angle`: The opening angle from the position of the play towards the goal posts must be large enough or otherwise the ball has to be kicked particularly forcefully.

4. *play.extrapolated_position_on_goalline* < `max_lateral_deviation`: The play must have been aimed sufficiently close towards the goal.

5. *play.receiver* is None **and** *play.target.distance_to_goalline* < `max_target_dist_to_ goalline` **or** *play.initial_speed* > `min_speed_general` **or** *play.receiver* is opposition

**Table 1. Features used for the pass/shot classification.**

| Feature | Type | Rationale |
|---|---|---|
| Distance from end position to opposition goal line | float | Shots are likely to cross the goal line |
| Distance to opposition goal | float | Shots are likely to be taken close to the opposition goal |
| Initial speed | float | Shots tend to be kicked more forcefully than passes |
| Receiver is opposition goalkeeper | boolean | Shots are often caught by the opposition goalkeeper. |
| Receiver is opposition field player | boolean | Shots are often blocked by opponents |
| Has receiver | boolean | Shots are not intended to be received by a player. |
| Goal angle | float | Shots are rarely taken from acute angles |
| Extrapolated lateral deviation | float | Shots are usually aimed in the direction of the goal rather than away from it. |
| Angle to goal | float | Shots are usually aimed in the direction of the goal rather than away from it. |
| Direct play or deflection follows | boolean | Shots are often blocked or deflected. |
| Progressed distance towards goal | float | Shots are usually taken towards the goal. |
| Distance to closest opponent | float | Shots are often taken under pressure. |

goalkeeper *play.initial_speed* > `min_speed_gk`: A shot must either end at the goalline or must be kicked forcefully enough. The required speed threshold differs depending on whether the ball hits an outfield player or the opposition goalkeeper.

**Machine learning.** To automatically learn rulesets with various complexity, we also evaluate decision trees with a fixed number of leaves. The structure of the decision trees is optimized by fitting other hyperparamters to data.

Additionally, we use different black box machine learning models to estimate whether and how much additional structure in the data can be uncovered when human-understandable rules are not required. These models are a random forest, a SVM and AdaBoost with decision trees as base classifier.

**Baseline.** Baseline performance is measured using a dummy predictor that always predicts the most frequent class, i.e. "Pass" in the training data.

## 4. Evaluation

### Data sets

We use positional and event data from four different providers for the evaluation.

- `Metrica` [17]: Anonymized sample data published by *Metrica Sports* consisting of 3 games with synchronized positional and event data.

- `Stats` [18]: Synchronized positional and event data of consecutive games of a professional men's national team in various competitions, provided by *Stats Perform*, 14 games.

- `Euro` [19]: Positional data from the men's European Championship 2021, provided by *Tracab*, complemented with independent *Hudl Sportscode* event data, 4 games.

- `Subsequent` [20]: Synchronized positional and event data of consecutive games of a professional men's national team in various competitions, provided by *Subsequent*. 6 games.

The positional data from all four providers was collected using optical tracking. *Tracab* and *Stats Perform* use in-venue camera systems whereas *Metrica* and *Subsequent* generate positional data from a single video recording and are therefore expected to be of lower quality. The positional data contains the x-y coordinates of the players and the ball during the match, captured at 25 Hz (`Metrica`, `Euro`, `Subsequent`) and 10 Hz (`Stats`) respectively. Due to the nature of the data, the event information contained in the four data sets is heterogeneous. Nevertheless, all four data sets do record passes and shots, including a timestamp which can be used to synchronize the respective action with the positional data. Additional information that is included in all data sets is the identity of the player who performed the pass or shot. An indication about the success of the pass as well as the identity of the pass receiver and the location at which the pass or shot starts and ends is not present in all data sets. The success of a pass and the identity of its receiver can however be deduced from the information given about the next ball-related event after the pass.

From qualitative inspection, it is obvious that the bundled positional and event data have not been generated independently from each other. For example, in the `Metrica` data set, the position of the ball is typically exactly equal to the position of the player who is currently in possession of the ball—a phenomenon that has also been observed in previous studies on data from other providers [5]. This observation strongly suggests that the position of the ball has been partly or even entirely reconstructed from manually annotated events. To a lesser degree, such artifacts are also apparent in the `Stats` and `Subsequent` datasets, but not in the in-venue positional data from *Tracab* within the `Euro` data set.

In the `Euro` dataset, the event data was obtained from the online platform *Wyscout*. However, the timestamps of the events were not accurately aligned with the actual events. For that reason, we corrected the timestamps of all passes, shots and game interruptions manually using broadcast footage of the games. This process also involved some minor corrections to the data, for example for events that were clearly missing or duplicate. Around 3 percent of events have been added or removed for such reasons. No positional data was used during this process.

## Game segmentation

Since detected events have to be matched with reference events, the validation routine needs to operate on contiguous segments of play in which to search for matching events. Since our models contain parameters to be fitted, we need at least two such segments in order to obtain a training set and a test set. Naturally, the data could be divided into games or halves. But since our smallest data set contains only 6 halfs, a subdivision along halfs would be too coarse to obtain a representative test set.

More blended data sets are obtained by instead dividing the game at any sufficiently long interruption. The game interruptions must be long enough so that a detected event and its true corresponding reference event almost certainly cannot end up in different segments. A higher interruption length therefore minimizes the risk of unwanted separations while a shorter interruption length increases the number of available segments. We found a minimum interruption time of 2 seconds to be a good compromise.

## Temporal matching

To determine whether a detected event matches a reference event, they have to be temporally matched. Since we treat passes and shots as composed of two atomic events which are modeled without a duration (hits and game interruptions), it is sensible to match two plays by individually matching their two constituent events. A play is matched if both of its consitutient atomic events match a detected event. The atomic events match if they are no further than a certain time span (matching window) apart from each other.

The choice of the optimal matching window involves the following trade-off: If the matching window is too small, it mistakenly misses actually matching events and underestimates the performance of the algorithm. If it is too large, it could mistakenly match unrelated plays and overestimate the performance of the algorithm. Therefore, additional information like the player and the location of the play should be used to establish truthful matching conditions. The shots and passes in our data sets share only one additional variable: the player who took the play. Therefore, we further require that this player must be equal for two events to be matched.

The dependency of detection performance on the choice of matching window is depicted in Fig 1. We qualitatively estimate the optimal matching window as 500 milliseconds for `Stats` [18], `Metrica` [17], and `Subsequent` [20], and 1000 milliseconds for `Euro` [19]. This is roughly where the scores begin to increase much slower than before, which indicates that the majority of actually corresponding events have been matched.

Any ambiguities where an event matches multiple candidates are resolved by finding a maximum cardinality matching for each segment using the Hopcroft-Karp algorithm.

## Choice of evaluation metrics

**Play detection.**   The relevant raw performance metrics for play detection are as follows:

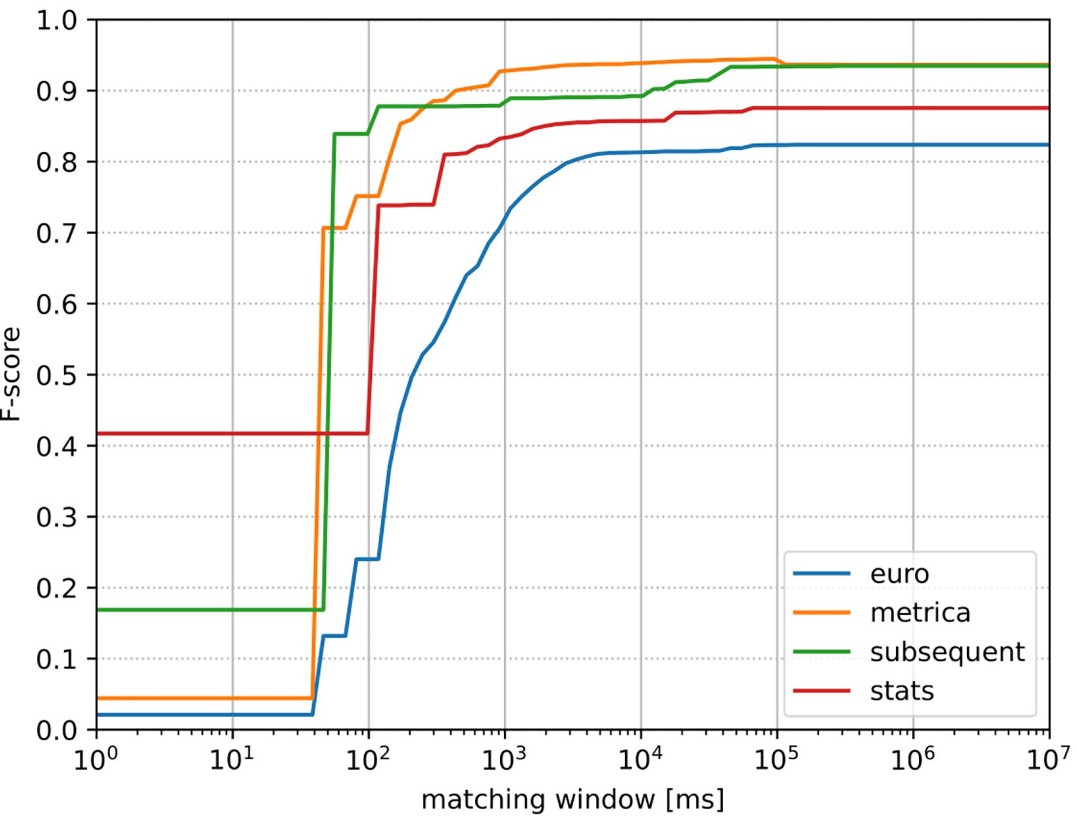

**Fig 1. Relationship between matching window and F-score in the training data.**

- Precision $P_{play} = \frac{\text{\# detected plays matched with a reference pass or shot}}{\text{\# detected plays}}$

- Pass Recall $R_{play,pass} = \frac{\text{\# detected plays matched with a reference pass}}{\text{\# reference passes}}$

- Shot Recall $R_{play,shot} = \frac{\text{\# detected plays matched with a reference shot}}{\text{\# reference shots}}$

Passes and shots constitute a class imbalance, as passes are about 40 times more common than shots in football. Since different categories of events are typically of separate interest in analysis rather than being mixed together, it is most appropriate to assign equal importance to passes and shots as categories, i.e. to assign more weight to an individual shot than an individual pass. This way, the algorithm will be optimized in a way that allows it to be used for the analysis of both types of events rather than being optimized to recognize mostly passes.

Based on that line of reasoning, we compute the macro-averaged recall $R_{play}$.

$$R_{play} = \frac{R_{play,pass} + R_{play,shot}}{2} \tag{3}$$

$R_{play}$ is then used to compute the F1-score $F_{play}$ that serves as the optimization target to balance overall recall and precision:

$$F_{play} = 2\frac{R_{play} \cdot P_{play}}{R_{play} + P_{play}} = \frac{2P_{play} \cdot (R_{play,pass} + R_{play,shot})}{2P_{play} + R_{play,shot} + R_{play,pass}} \tag{4}$$

**Pass/shot classification.** The classification into passes and shots can be evaluated independently of the preceding play detection step using the precision and recall of passes and shots relative to the set of successfully matched plays.

- Shot Precision $P_{shot} = \frac{\text{\# classified shots matched with a reference shot}}{\text{\# classified shots matched with a reference shot or pass}}$

- Pass Precision $P_{pass} = \frac{\text{\# classified passes matched with a reference pass}}{\text{\# classified passes matched with a reference shot or pass}}$

- Shot Recall $R_{shot} = \frac{\text{\# classified shots matched with a reference shot}}{\text{\# detected shots and passes matched with a reference shot}}$

- Pass Recall $R_{shot} = \frac{\text{\# classified passes matched with a reference pass}}{\text{\# detected shots and passes matched with a reference pass}}$

Again, the optimization target must account for class imbalance. In this case, since precision and recall are available for both classes, we can use the macro-average of the two regular F1-scores to obtain our optimization target $F_{avg}$.

- $F_{shot} = \frac{R_{shot} \cdot P_{shot}}{R_{shot} + P_{shot}}$

- $F_{pass} = \frac{R_{pass} \cdot P_{pass}}{R_{pass} + P_{pass}}$

- $F_{avg} = \frac{F_{shot} + F_{pass}}{2}$

To quantify the overall performance of the classifier, we also report variants of the above metrics relative to the total number of reference and detected events, respectively.

- $P'_{shot} = \frac{\text{\# correctly classified shots}}{\text{\# classified shots (including among falsely detected plays)}}$

- $P'_{pass} = \frac{\text{\# correctly classified passes}}{\text{\# classified passes (including among falsely detected plays)}}$

- $R'_{shot} = \frac{\text{\# correctly classified shots}}{\text{\# reference shots}}$

- $R'_{pass} = \frac{\text{\# correctly classified passes}}{\text{\# reference passes}}$

- $F'_{shot} = 2\frac{P'_{shot} \cdot R'_{shot}}{P'_{shot} + R'_{shot}}$

- $F'_{pass} = 2\frac{P'_{pass} \cdot R'_{pass}}{P'_{pass} + R'_{pass}}$

- $F'_{avg} = \frac{F'_{pass} + F'_{shot}}{2}$

## Parameter optimization

Each data set is split into a training set and a test set with a 65-35 ratio of game segments. The resulting number of shots and passes is shown in Table 2.

**Table 2. Overview of training and test sets.**

| Dataset | Games | Training passes | Test passes | Training shots | Test shots |
|---|---|---|---|---|---|
| Stats [18] | 14 | 9831 | 5053 | 212 | 117 |
| Euro [19] | 4 | 2813 | 1495 | 62 | 27 |
| Metrica [17] | 3 | 2422 | 1277 | 48 | 20 |
| Subsequent [20] | 6 | 4364 | 2375 | 105 | 51 |

The parameters of the play detector are fitted on the entire training set using 300 iterations of Bayesian optimization within the following bounds.

$$\texttt{min\_vicinity} \in [0.01m, 10m]$$

$$\texttt{min\_acc} \in \left[0\frac{m}{s^2}, 120\frac{m}{s^2}\right]$$

$$\texttt{max\_deflection\_time} \in [0ms, 1000ms]$$

Similarly, the 8 parameters of the manual rules classifier are fitted using Bayesian optimization with 120 iterations and the following bounds.

$$\texttt{min\_progression} \in [-100m, 50m]$$

$$\texttt{max\_dist\_to\_goal} \in [0m, 50m]$$

$$\texttt{min\_opening\_angle} \in [0, 180°]$$

$$\texttt{max\_lateral\_deviation} \in [0m, 34m]$$

$$\texttt{max\_target\_dist\_to\_goalline} \in [0m, 10m]$$

$$\texttt{min\_speed\_general} \in \left[0\frac{m}{s}, 50\frac{m}{s}\right]$$

$$\texttt{min\_speed\_gk} \in \left[0\frac{m}{s}, 100\frac{m}{s}\right]$$

$$\texttt{min\_speed\_from\_bad\_angle} \in \left[0\frac{m}{s}, 100\frac{m}{s}\right]$$

The hyperparameters of the machine learning models are fitted using a 10 times repeated 10-fold stratified cross-validation on the training set using 250 iterations of Bayesian parameter search. For the decision trees, the parameter max_leaves is instead fixed to various values.

## 5. Results

### Play detection

As shown in Table 3, play detection performs well for the Stats, Subsequent and Metrica data sets, achieving macro-averaged F-scores of 0.87, 0.88, and 0.83 respectively. The more realistic Euro data set, where positional and event data are decoupled, achieves a significantly weaker score of 0.70. Shots display a lower class-specific recall than passes across all data sets.

**Table 3. Evaluation results for play detection.**

| Dataset | Precision $P_{play}$ | Recall $R_{play,pass}$ | Recall $R_{play,shot}$ | F-score $F_{play}$ |
|---|---|---|---|---|
| Stats [18] | 0.84 | 0.91 | 0.88 | 0.87 |
| Euro [19] | 0.63 | 0.82 | 0.72 | 0.70 |
| Metrica [17] | 0.89 | 0.90 | 0.70 | 0.83 |
| Subsequent [20] | 0.89 | 0.96 | 0.80 | 0.88 |

**Table 4. Optimized parameter values for play detection.**

| Dataset | min_acc $[\frac{m}{s^2}]$ | min_vicinity [$m$] | max_deflection_time [$ms$] |
|---|---|---|---|
| Stats [18] | 25.9 | 3.0 | 748 |
| Euro [19] | 63.3 | 1.5 | 99 |
| Metrica [17] | 26.8 | 1.4 | 91 |
| Subsequent [20] | 9.5 | 3.0 | 487 |

As can be seen from Table 4, the optimized values for min_acc, min_vicinity, and max_deflection_time vary significantly between the data sets.

## Pass/shot classification

The results of the pass and shot classifier are shown in Fig 2.

Due to the strong imbalance of the data, the baseline model, which always predicts the majority class, yields a macro average F-score of roughly 0.5. All classifiers easily outperform this baseline.

AdaBoost and Random Forest show the strongest performance with F-scores $F_{avg}$ ranging from 0.93 to 0.95 for Stats, Euro and Metrica, and 0.85 to 0.87 for Subsequent. The performance of the rule-based classifiers is almost as strong with F-scores between 0.83 and 0.91.

Fig 3 shows the performance of the decision trees depending on the fixed maximum number of leaves. The performance converges already after 3-6 leaves, after which the possibility to add more splitting rules does not lead to a clear performance improvement.

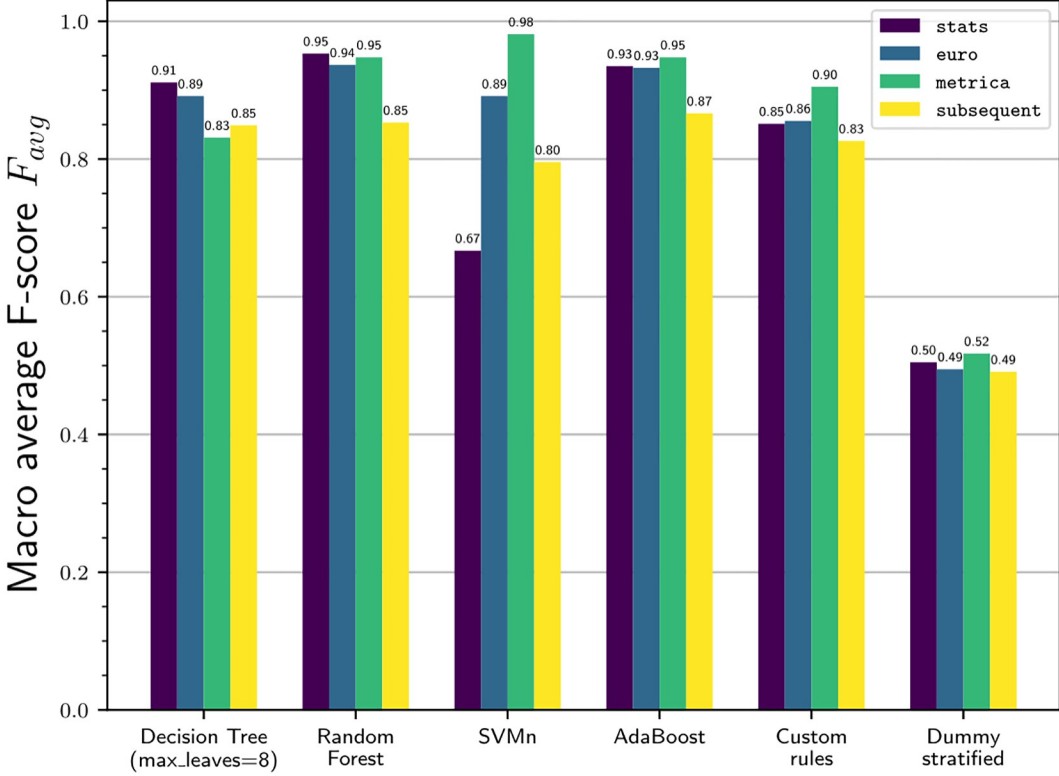

**Fig 2. F-scores of pass/shot classifier relative to the correctly detected plays.**

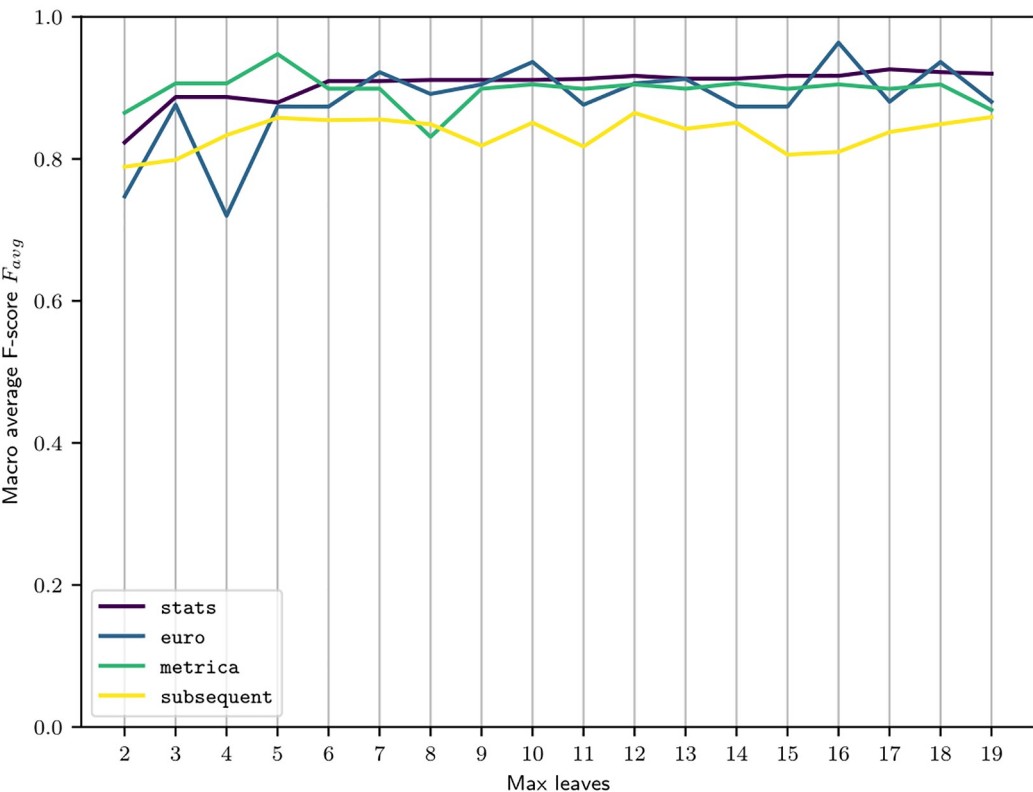

**Fig 3. Performance of decision trees by number of leaves.**

General insights into feature importance are drawn by inspecting the decision trees and the random forest. As shown in Table 5, the 2-leaf trees either use the rule "`Distance End Position to Goalline` $< X$" where $X$ is around 2-5 meters away from the goal line or "`Distance Start Position to Goal` $< X$" where $X$ is around 20-30 meters to identify shots. There are only two other rules used in the decision trees up to 4 leaves, namely "`Opening angle` $< 12.4°$" and "`Lateral end position (projected)` $< X$" with $X$ between 5 and 10m. Beginning with the third split, the decision trees begin to learn redundant splits, assigning the same class to both child nodes. Inspecting the impurity-based feature importance of the random forest (Fig 4) confirms the paramount role of these four features in classification, while some of the remaining features such as the initial speed of the ball, the distance of the closest attacker to the goal and the progressed distance towards the goal also appear to be relevant.

**Table 5. The logical rules learned by the first three decision trees to classify a play as a shot, for each data set.**
$D_{start,goal}$: Distance from play origin to goal. $D_{start,goal}$: Distance from play end position to goal-line. $A_{open}$: Opening angle of the goal from play origin. $Y_{end*}$: End position of the play, projected onto goal-line.

| Data set | 2 Leaves | 3 Leaves | 4 Leaves |
|---|---|---|---|
| `Euro` [19] | $D_{start,goal} < 30.1m$ | $A_{open} > 12.4°$ **and** $Y_{end*} < 9.48m$ | $D_{end,gl} < 2.14m$ **and** $A_{open} > 12.1°$ |
| `Stats` [18] | $D_{end,gl} < 3.08m$ | $D_{end,gl} < 3.08m$ **and** $Y_{end*} < 8.15m$ | $A_{open} > 12.6°$ **and** $Y_{end*} < 7.16m$ |
| `Metrica` [17] | $D_{end,gl} < 2.46m$ | $D_{end,gl} < 4.18m$ **and** $A_{open} > 12.9°$ | $D_{end,gl} < 2.46m$ **and** $A_{open} > 10.9°$ |
| `Subsequent` [20] | $D_{end,gl} < 3.91m$ | $D_{end,gl} < 3.91m$ **and** $A_{open} > 10.4°$ | $D_{end,gl} < 3.91m$ **and** $A_{open} > 10.4°$ |

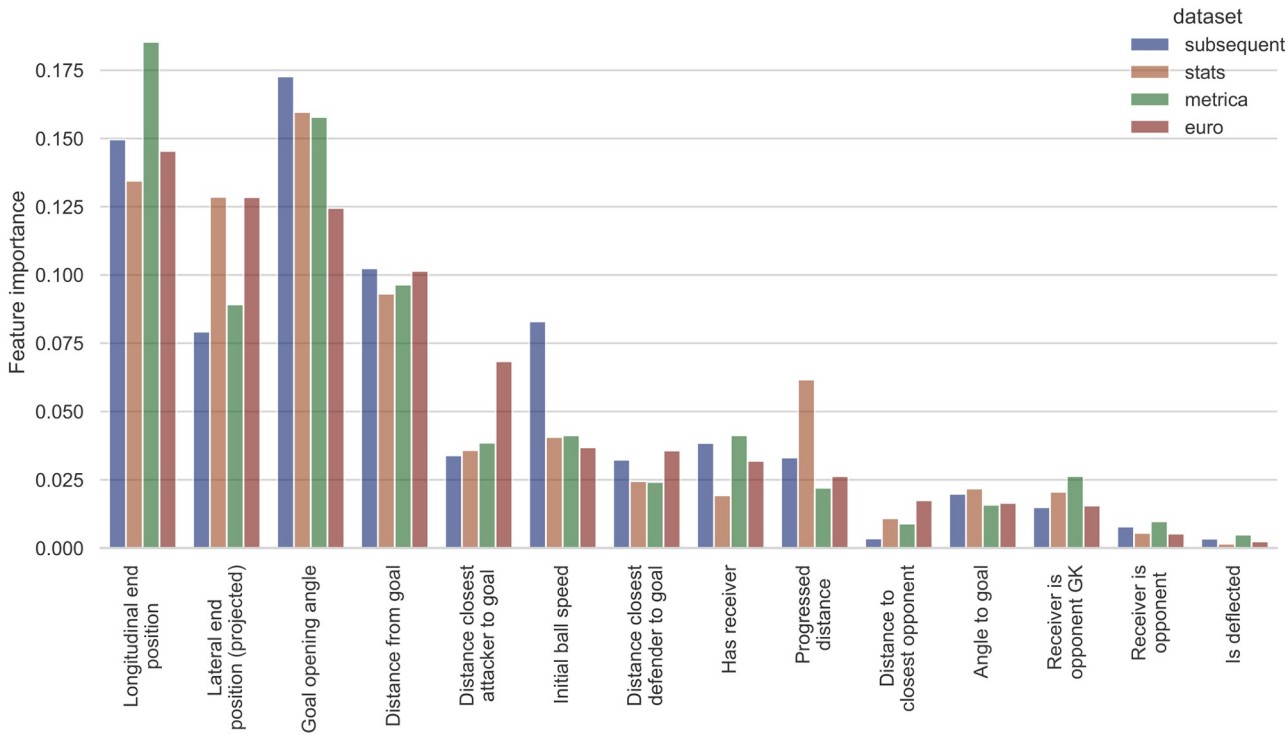

**Fig 4. Impurity-based feature importance for random forest.**

The combined performance of the event detection routine is shown in Table 6, using Ada-Boost for shot classification. The total macro-averaged F-score for detecting passes and shots range from 0.67 to 0.82, depending on the data set. As is also evident from the evaluation of the detector alone (Table 3), shots achieve much lower scores than passes. Passes are detected with an overall F-score of around 0.9, except for the Euro dataset which achieves a lower score of 0.71.

## 6. Discussion

Our results show that the performance of pass and shot detection is heavily dependent on the characteristics of the data set. Regarding the data sets Subsequent, Metrica and Stats, our study reproduces the previously observed F-scores in pass detection, while using a minimalistic detection algorithm. Our scores between 0.87 and 0.92 for those data sets are in line with the results from Morra et al. (0.89) [6], Khaustov and Mozgovoy (0.86 unsuccessful passes, 0.93 successful passes) [7], Richly et al. (0.89) [16], and Vidal-Codina et al. (0.92) [5].

**Table 6. Total classification performance of play detector + AdaBoost shot classifier.**

| Dataset | $P'_{pass}$ | $R'_{pass}$ | $F'_{pass}$ | $P'_{shot}$ | $R'_{shot}$ | $F'_{shot}$ | $F'_{avg}$ |
|---|---|---|---|---|---|---|---|
| Stats [18] | 0.84 | 0.91 | 0.87 | 0.78 | 0.76 | 0.77 | 0.82 |
| Euro [19] | 0.63 | 0.82 | 0.71 | 0.65 | 0.59 | 0.62 | 0.67 |
| Metrica [17] | 0.89 | 0.90 | 0.89 | 0.65 | 0.65 | 0.65 | 0.77 |
| Subsequent [20] | 0.88 | 0.96 | 0.92 | 0.61 | 0.55 | 0.58 | 0.75 |

The large differences of the optimal parameter values (Table 4) indiciate that the utilized data sets are heterogeneous. The large difference in the optimal acceleration threshold stems from the acceleration being manually computed as the second derivative of the position. Therefore, the particularly high optimal threshold for the `Euro` data set indicates that its positional data is indeed the most raw among the four providers.

In contrast, `Subsequent`, `Stats` and `Metrica` likely used event data to post-process their positional data. Therefore, the performance of the pass and shot detection algorithm on these data sets is likely an overestimation of its true ability to identify these plays in raw positional data. Its performance on the `Euro` data set (0.71 for passes, 0.62 for shots) is a more truthful reflection of its capabilities as the positional and event data within this data set are independent and the positional data appears to contain few event artifacts.

Given these results and assuming that other algorithms would experience a similar drop in performance when evaluated on independent data (see the results of Vidal-Codina et al. [5] for a rough impression), the current state of the art in detecting events from positional data seems unsatisfying. One third of the detected passes or shots would not appear in the manual event data that analysts are used to, and conversely, around one third of the manually collected events would be missing. Even when factoring in the inherent subjectivity of manual event data, this appears to be a troubling deficit in accuracy.

Our two-step event detection pipeline exposes play detection rather than the subsequent classification as the primary issue. Qualitative post-hoc inspection of the detector's mistakes on the `Euro` data set reveals the following causes of suboptimal performance:

- Inaccuracies of the positional data: For example, the ball position in the data from *Tracab* comes with small artificats where the ball sometimes changes its velocity abruptly during a pass. This is falsely recognized as a hit if some player happens to be close by, for example when a pass goes slightly past or over a player. The algorithm can account for that by reducing the required player-ball-distance to determine a hit. However, the required player-ball-distance also needs to be large enough to account for the reach of the player and noise in the ball and player positions. The algorithm cannot account for both at the same time.

- The algorithm struggles when many players are located around the ball and the ball is accelerated either through dribbling or artifacts. In these situations, the closest player to the ball can change frequently without an actual possession change. This effect is much less prevalent in the other data sets because the ball "sticks" to the currently possessing player as presumably determined from manually collected event information.

- The lack of ball height data makes it difficult to identify irrelevant x-y-accelerations due to bouncing.

- Errors in the reference data, in particular missing events.

Shot classification on the other hand performs well across all data sets. Given the quick convergence of the decision trees, it seems that for most data sets, one to three human-understandable rules are already sufficient to differentiate shots from passes with an accuracy of around 90%. These rules primarily operate upon the start and end position relative to the opponent's goal. At least a small additional boost in accuracy can be achieved using machine learning. A small set of rules is therefore sufficient for differentiating shots from passes and can be used as a more objective definition for this kind of event.

Also, like other rule-based methods, the proposed algorithm runs in linear time, which makes it suitable for real-time application which is an essential requirement in the industry where data must be streamed to clients during matches.

## 7. Conclusions and future work

We proposed an evaluation routine for event detection in football that deals with the temporal, hierarchical, and imbalanced nature of the task. It demonstrates solutions for the problems that this type of classification task poses, like temporal matching and the choice of evaluation metrics. It can be applied to football event detection as well as related tasks like object detection and sentiment analysis.

As evaluated by the novel validation routine, the proposed two-step event detection algorithm effectively detects passes and shots from a stream of positional data, reaching state of the art performance in the majority of examined data sets while using a rule-based algorithm with minimal complexity. Using a small set of rule leads to an easily interpretable and extendable definition of the detected events which is an essential requirement to improve the objectivity and accuracy of further insights gained by researchers and practitioners.

For the most realistic data set examined, the detection of plays from raw positional data proved as the main obstacle in achieving high-accuracy results. Further analysis suggests that this problem could be partially mitigated in the future by richer and more accurate positional data. However, once plays have been detected, the task of differentiating passes and shots is relatively simple and can be performed using a low number of human-understandable rules. This is a promising insight to help put event-based performance analysis of passes and shots onto a more reliable foundation.

While rule-based pass and shot detection on positional data appears to achieve high accuracy given the current state of the art, our study found that this impression is most likely distorted by the fact that information from the reference data commonly spills over into the input data of the models. More research that performs high-quality evaluations on suitable algorithms in a realistic data setting is needed to determine viable solutions for the automation of manual event data collection.

While we provide a broad perspective on event detection performance through using four data sets from different providers, a limitation of this study is that the data sets themselves are relatively small, especially regarding the number of shots which are a much rarer event in football than passes. Also, our study focuses only on shots and passes while more complex and subjective events like tackles and dribbles and events like yellow cards which are virtually impossible to define in terms of spatio-temporal interactions would presumably achieve a far lower accuracy. For that reason, it seems hard to imagine rule-based event detection as a full-blown solution to the automation of event detection processes. However, it can serve as a complement to more comprehensive detection systems, especially for applications where flexibility, interpretability, and objectivity are paramount, such as academic studies or when existing game models of football clubs and federations need to be accommodated.

## Acknowledgments

Thanks to Philipp Schmid for his diligent assistance with the timestamp correction.

## Author Contributions

**Conceptualization:** Jonas Bischofberger.

**Formal analysis:** Jonas Bischofberger.

**Methodology:** Jonas Bischofberger.

**Project administration:** Jonas Bischofberger.

**Software:** Jonas Bischofberger.

**Supervision:** Arnold Baca, Erich Schikuta.

**Validation:** Jonas Bischofberger.

**Visualization:** Jonas Bischofberger.

**Writing – original draft:** Jonas Bischofberger.

**Writing – review & editing:** Jonas Bischofberger, Arnold Baca, Erich Schikuta.

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
