## [Decision Letter · Decision Letter 0]

26 Oct 2023

PONE-D-23-27502Event detection in football: Improving the reliability of match analysisPLOS ONE

Dear Dr. Bischofberger,

Thank you for submitting your manuscript to PLOS ONE. After careful consideration, we feel that it has merit but does not fully meet PLOS ONE’s publication criteria as it currently stands. Therefore, we invite you to submit a revised version of the manuscript that addresses the points raised during the review process.

We look forward to receiving your revised manuscript.

Kind regards,

Ersan Arslan, Ph.D.

Academic Editor

PLOS ONE

Journal Requirements:

4.  We note you have included a table to which you do not refer in the text of your manuscript. Please ensure that you refer to Table 5 in your text; if accepted, production will need this reference to link the reader to the Table.

**Additional Editor Comments:**

ACADEMIC EDITOR:

Dear corresponding Author, the Reviewers found some concerns with your manuscript. Please take care of their suggestion and send back the manuscript as soon as possibile. 

Reviewers' comments:

Reviewer's Responses to Questions

**Comments to the Author**

1. Is the manuscript technically sound, and do the data support the conclusions?

Reviewer #1: Yes

Reviewer #2: Yes

2. Has the statistical analysis been performed appropriately and rigorously? 

Reviewer #1: Yes

Reviewer #2: Yes

3. Have the authors made all data underlying the findings in their manuscript fully available?

Reviewer #1: Yes

Reviewer #2: Yes

4. Is the manuscript presented in an intelligible fashion and written in standard English?

Reviewer #1: Yes

Reviewer #2: No

5. Review Comments to the Author

Reviewer #1: Include the problem statement, objective, motivation, and paper organization in the introduction section.

List out the strengths, limitations, and techniques of recent related existing methods in the related work section.

Provide a detailed analysis of the datasets with their data details, feature details, and volume of datasets.

This work is not compared with any existing methods. The authors should provide any statistical, mathematical, or comparative proof to claim that this work is state-of-the-art.

Include findings, strengths, limitations, application area, and future work of the proposed method in the conclusion section.

Reviewer #2: Review Comments

Presented paper emphasizes existing research by validating a two-step rule-based pass and shot detection algorithm on four different data sets using a comprehensive validation routine that accounts for the temporal, hierarchical and imbalanced nature of the task. Our evaluation shows that pass and shot detection performance is highly dependent on the specifics of the data set. However, the following suggestions can be considered by the authors to further improve the quality of the manuscript.

I have some corrections and suggestions below:-

1. Authors must show explain the novel contribution of the work with proper justification of the outcomes. What novelty is established in this work compared to existing works?

2. The computational complexity in terms of time and space must be discussed. Also, compare the proposed method in terms of computational complexity?

3. Literature survey need to be updated based on current state of art methods. Some more paper based on Event detection in football:.

4. The abstract need to be improved and the outcome of the work in terms of achieved various other performance calculations must be included in the abstract.

5. Explaining the problem and the gaps in existing literature in a concise but self-contained way (although readers might wish to consult references, they should not be forced to do so)

6. Organization of the paper can be added at the end of introductions.

7. Comparative analysis of various performance parameters with respect to sate of art methods must be discussed. More recent state-of-the-art approaches should be compared; the experiments should use more sizable real-world data sets from public repositories (if any);

8. Add industrial significance of the proposed approach.

9. Results must be verified some more other data sets also. Describing in detail the data set used and what are the expected outcomes- widening the experimental comparison including other data and methods.

10. Comparative analysis of various performance parameters with respect to various data sets must be discussed. The comparison can be a bit unfair if different data is not used for comparative analysis.

11. Limitations of the proposed work must be included.

12. Precision vs. recall curves of the proposed algorithms with respect to different data sets must be included.

13. Implementation platforms with complete specifications of the system must be included.

14. How much data should be considered for training and testing for model implementation? Details of training and testing data sets must be tabulated.

14. To make the proposed algorithm of this article more readable use pseudo-code.

15. In all results tables’/figures utilized datasets like in table 2, 3 and 4 etc. must be cited with proper and specific citations.

16. Various visualized results based on proposed work must be added and also compared the results with existing work.

17. Comparative analysis with respect to real-time time analysis is missing?

6. PLOS authors have the option to publish the peer review history of their article (what does this mean?). If published, this will include your full peer review and any attached files.

Reviewer #1: **Yes: **Dr. Mahendra Prasad

Reviewer #2: No

---

## [Author Response · Author response to Decision Letter 0]

28 Dec 2023

Dear editors, dear reviewers,

I wish to express my sincere thanks to you for taking the time and consideration to review our work. The review process has raised valid issues which we tried to address and improve our manuscript accordingly. Below is our response to each of the points that have been raised.

Include the problem statement, objective, motivation, and paper organization in the introduction section.

- Extended the introduction accordingly.

List out the strengths, limitations, and techniques of recent related existing methods in the related work section.

- Added detail about the methods employed in previous studies.

Provide a detailed analysis of the datasets with their data details, feature details, and volume of datasets.

- Added details about the data sets, in particular details about relevant features and the number of relevant events in the training and test sets.

This work is not compared with any existing methods. The authors should provide any statistical, mathematical, or comparative proof to claim that this work is state-of-the-art.

- Added a discussion of time and space complexity, and a more explicit comparison of detection performance with previous results.

Include findings, strengths, limitations, application area, and future work of the proposed method in the conclusion section.

- Added details to the conclusion section.

1. Authors must show explain the novel contribution of the work with proper justification of the outcomes. What novelty is established in this work compared to existing works?

- Added some clarity concerning our contributions throughout the paper. The novel contributions have been listed at the end of the introduction.

2. The computational complexity in terms of time and space must be discussed. Also, compare the proposed method in terms of computational complexity? 

- Added time and space complexity. All rule-based event detection algorithms should trivially be of linear complexity as events are tightly localized within in the positional data.

3. Literature survey need to be updated based on current state of art methods. Some more paper based on Event detection in football:. 

- We decided to limit ourselves to studies that operate upon positional data and believe our selection of such studies should be complete. In the case that we missed scientific work on that specific topic, we would be happy to learn about it and add it to the paper! We did not explicitly include papers where events are detected directly from video due to the large categorical differences between those two approaches, especially regarding their main goals and areas of application (comprehensive industry-scale automation vs. research purposes and internal model building in clubs/federations), as touched upon in the introduction section. A paragraph about the practical role of the proposed algorithm has been added in the conclusion.

4. The abstract need to be improved and the outcome of the work in terms of achieved various other performance calculations must be included in the abstract. 

- Added more performance calculations in the abstract, mentioning the outcomes more explicitly.

5. Explaining the problem and the gaps in existing literature in a concise but self-contained way (although readers might wish to consult references, they should not be forced to do so) 

- Added more details to the literature background and pointed out the research gap more clearly.

6. Organization of the paper can be added at the end of introductions. 

- Added a corresponding paragraph.

7. Comparative analysis of various performance parameters with respect to sate of art methods must be discussed. More recent state-of-the-art approaches should be compared; the experiments should use more sizable real-world data sets from public repositories (if any); 

- Added a more explicit comparison of performance parameters with the state of the art and mentioned the size of the data sets as a limitation. While larger data sets would certainly be beneficial, those are not available to us due to the proprietary nature of most football match data. I also added a table with an overview of the number of reference events for each of the four data sets to provide a more explicit impression of our sample sizes.

8. Add industrial significance of the proposed approach. 

- Added a paragraph in the conclusion section.

9. Results must be verified some more other data sets also. Describing in detail the data set used and what are the expected outcomes- widening the experimental comparison including other data and methods. 

- Added detail the data used. More data is not available due to the proprietary nature of football match data.

10. Comparative analysis of various performance parameters with respect to various data sets must be discussed. The comparison can be a bit unfair if different data is not used for comparative analysis. 

- Completed the comparison such that each comment in the results section includes all the analysed data sets without leaving out any single one.

11. Limitations of the proposed work must be included. 

- Added a paragraph in the conclusion discussing limitations.

12. Precision vs. recall curves of the proposed algorithms with respect to different data sets must be included. 

- Since not all classifiers are probabilistic, we decided to use F1-scores to allow for a comparison between classifiers.

13. Implementation platforms with complete specifications of the system must be included. 

- Added information about the software framework used for implementation. Hardware specifics would not be informative since the code is executable on any regular computer with a compatible Python environment and sufficient RAM.

14. How much data should be considered for training and testing for model implementation? Details of training and testing data sets must be tabulated. 

- Added more information about the training and test sets in tabular form.

14. To make the proposed algorithm of this article more readable use pseudo-code. 

- Added pseudo-code.

15. In all results tables’/figures utilized datasets like in table 2, 3 and 4 etc. must be cited with proper and specific citations. 

- Added the suggested citations in all tables.

16. Various visualized results based on proposed work must be added and also compared the results with existing work. 

- Added a more explicit performance comparison in the discussion section.

17. Comparative analysis with respect to real-time time analysis is missing? 

- Added a corresponding part in the discussion section.

If there remain any issues with our revised manuscript, we would be happy to address those!

Best regards

Jonas Bischofberger, corresponding author

---

## [Decision Letter · Decision Letter 1]

20 Jan 2024

Event detection in football: Improving the reliability of match analysis

PONE-D-23-27502R1

Dear Dr. Bischofberger,

We’re pleased to inform you that your manuscript has been judged scientifically suitable for publication and will be formally accepted for publication once it meets all outstanding technical requirements.

Kind regards,

Ersan Arslan, Ph.D.

Academic Editor

PLOS ONE

Additional Editor Comments (optional):

Reviewers' comments:

Reviewer's Responses to Questions

**Comments to the Author**

1. If the authors have adequately addressed your comments raised in a previous round of review and you feel that this manuscript is now acceptable for publication, you may indicate that here to bypass the “Comments to the Author” section, enter your conflict of interest statement in the “Confidential to Editor” section, and submit your "Accept" recommendation.

Reviewer #1: All comments have been addressed

Reviewer #2: All comments have been addressed

2. Is the manuscript technically sound, and do the data support the conclusions?

Reviewer #1: Yes

Reviewer #2: Yes

3. Has the statistical analysis been performed appropriately and rigorously? 

Reviewer #1: Yes

Reviewer #2: Yes

4. Have the authors made all data underlying the findings in their manuscript fully available?

Reviewer #1: Yes

Reviewer #2: Yes

5. Is the manuscript presented in an intelligible fashion and written in standard English?

Reviewer #1: Yes

Reviewer #2: Yes

6. Review Comments to the Author

Reviewer #1: The authors have addressed the raised queries and the authors have improved the manuscript. I do not have more queries related to this manuscript.

Reviewer #2: All my comments has been added and successfully modified based on comments. I accept in current form.

7. PLOS authors have the option to publish the peer review history of their article (what does this mean?). If published, this will include your full peer review and any attached files.

Reviewer #1: **Yes: **Dr. Mahendra Prasad

Reviewer #2: **Yes: **MOHAMMAD FARUKH HASHMI

---

## [Editor Report · Acceptance letter]

21 Mar 2024

PONE-D-23-27502R1 

PLOS ONE

Dear Dr. Bischofberger, 

I'm pleased to inform you that your manuscript has been deemed suitable for publication in PLOS ONE. Congratulations! Your manuscript is now being handed over to our production team.

Kind regards, 

on behalf of

Dr. Ersan Arslan 

Academic Editor

PLOS ONE